# A Shift in Communities of Conspicuous Macrocrustaceans Associated with Caribbean Coral Reefs following a Series of Environmental Stressors

Melissa K. Dubé [1], Cecilia Barradas-Ortiz [2], Fernando Negrete-Soto [2], Lorenzo Álvarez-Filip [2], Enrique Lozano-Álvarez [2] and Patricia Briones-Fourzán [2,*]

[1] Posgrado en Ciencias del Mar y Limnología, Universidad Nacional Autónoma de México, Av. Universidad 3000, Coyoacán, Mexico City 04510, Mexico; mel.kdube@gmail.com

[2] Unidad Académica de Sistemas Arrecifales, Instituto de Ciencias del Mar y Limnología, Universidad Nacional Autónoma de México, Puerto Morelos 77580, Mexico; barradas@cmarl.unam.mx (C.B.-O.); fnegrete@cmarl.unam.mx (F.N.-S.); lorenzo@cmarl.unam.mx (L.Á.-F.)

[*] Correspondence: briones@cmarl.unam.mx

**Abstract:** In 2015, the communities of reef-associated motile macrocrustaceans (decapods and stomatopods) were compared between two coral reefs with contrasting levels of degradation in Puerto Morelos (Mexican Caribbean), "Limones", less degraded, with a healthy live coral cover, and "Bonanza", more degraded, with less live coral and more macroalgae. Since then, several stressors have impacted Puerto Morelos. Massive influxes of floating Sargassum, which reached record levels in 2018, 2021, and 2022, have exacerbated the already high eutrophication of the reef lagoon. An outbreak and rapid propagation of the Stony Coral Tissue Loss Disease in 2018 changed the functionality of reefs. Three back-to-back hurricanes struck the coast close to Puerto Morelos in October 2020 and another one in August 2021. We repeated the study in 2022 to examine the potential changes in the habitat and communities of reef-associated crustaceans since 2015. Reef degradation did not increase significantly between 2015 and 2022, but crustacean species richness, diversity, evenness, and dominance, which differed between reefs in 2015, became similar between reefs in 2022, as did the crustacean community composition. The abundance of herbivore crabs increased in Limones, displacing the coral- and hydrocoral-mutualistic crabs and the abundance of detritivore hermit crabs increased in Bonanza. These results suggest a taxonomic homogenization between reefs, apparently related to subtle ecological changes not necessarily captured by standard metrics of reef condition.

**Keywords:** community composition; crustaceans; decapods; hurricanes; habitat degradation; taxonomic homogenization; ecological indices

## 1. Introduction

Tropical coral reefs are among the most diverse marine ecosystems and provide important ecosystem services, such as coastal protection, provision of fisheries resources, and tourist income, but are vulnerable to climate change and other anthropogenic stressors, such as a decrease in seawater quality due to nutrient inputs, destruction of coastal habitats, destructive fishing, and the introduction of non-native species [1,2]. Increased temperature and eutrophication favor outbreaks of diseases that kill reef-building corals, with a concomitant increase in macroalgal cover, resulting in the loss of structural complexity of reefs [3]. Caribbean coral reefs have some of the highest rates of degradation [4,5], potentially affecting the biological communities associated with these systems and altering trophic webs [6–8].

Over the last few decades, the northern Mexican Caribbean coast (state of Quintana Roo) has been subjected to expansive coastal development, massive tourist visitation (~15 million/year), and an exponential increase in the local human populations. Despite



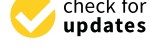

the increase in wastewater concomitant with this growth, sewage management and control remain poor because wastewater plants are scarce and only apply primary treatments to wastewater [8–11]. Due to the karstic nature of the region, groundwater discharges into reef lagoons through sediments and submarine springs; therefore, eutrophication, pollution, and sedimentation resulting from the construction of infrastructure and inadequate wastewater treatment are considered a major driver of declining reef condition in the region [8–11]. More recently, coral reefs in this region have been further affected by two unprecedented phenomena that appear to be related to large-scale ecological changes associated with global climate change: the massive influxes of floating Sargassum spp. macroalgae [12,13], and the Stony Coral Tissue Loss Disease (SCTLD), which affects over 20 species of hard corals [14].

The first massive influxes of pelagic *Sargassum*, which are driven from the so-called "great Atlantic *Sargassum* belt" into the Caribbean basin by sea currents [15], arrived in the Mexican Caribbean in 2014. Upon reaching the coastal areas, thousands of tons of Sargassum become stranded and rapidly decay, turning the coastal waters brown and turbid. These so-called "*Sargassum* brown tides" increase nutrient concentration in the water column and decrease light penetration, pH, and oxygen concentration [12]. Nearshore seagrasses are replaced by faster-growing macroalgae and epiphytes, and nearshore corals can suffer total or partial mortality [12]. Although the effects are more pronounced in the reef lagoons [11,12,16], they can propagate toward and affect the coral reefs [8]. Sargassum brown tides may cause mass mortalities of marine biota due to the low pH and oxygen concentration and high ammonium levels [17,18]. The floating *Sargassum* masses are also a potential vehicle for non-native fauna, raising concerns about their effects on local communities [19].

Since the 1980s, numerous diseases have been affecting stony corals, particularly acroporids, resulting in their loss, which, in conjunction with hurricanes, has substantially decreased reef structural complexity [3,20]. However, SCTLD, which attacks numerous coral species except for acroporids, was first observed in the upper Florida Keys, USA, in September 2014 [21]. An unprecedented outbreak of SCTLD in the Mexican Caribbean was reported in 2018. Unlike other coral diseases, SCTLD kills the coral colonies rapidly and spreads very quickly, having affected the entire Mexican Caribbean in only a few months, significantly changing the coral communities and functionality of the reefs [3,22,23].

Most studies addressing the emergence and propagation of SCTLD and the massive *Sargassum* influxes have focused on their effects on the habitat-forming species, i.e., stony corals and seagrasses, respectively. However, a large proportion of the biodiversity in coral reef systems is composed of motile invertebrates that span a broad size range and are important components of many trophic webs [24–26], and there is no understanding of how such recent phenomena may threaten this biodiversity [26,27]. Crustaceans constitute a large part of the reef-associated invertebrate fauna and, hence, have been a target taxon for these types of studies [28–30]. Interestingly, some studies have reported a decrease in reef-associated invertebrates, including crustaceans, with habitat degradation (e.g., [31]), whereas others have found either little difference (e.g., [26,32]) or an increase in diversity measures of this fauna with habitat degradation (e.g., [27,33–36]).

In the Puerto Morelos Reef National Park, located on the northern Mexican Caribbean coast, the communities of conspicuous macrocrustacean (herein defined as motile decapods and stomatopods larger than ~1.5 cm [35]) associated with two coral reefs of contrasting levels of degradation were studied in 2015 [35]. Species richness and abundance were higher on the more degraded reef, but specialists were more abundant on the less degraded reef. Given the ongoing Caribbean-wide tendency to an increase in coral reef degradation, González-Gómez et al. [35] stressed the importance of different types of microhabitats and the occurrence of mutualistic relationships for maintaining diversity and abundance of reef-associated macrocrustaceans. The study of González-Gómez et al. [35] was conducted a few years before the SCTLD outbreak and when the massive Sargassum influxes were only beginning. Furthermore, in October 2020, the northern Mexican Caribbean coast

was hit by three consecutive hurricanes, TS Gamma on October 3, H2 Delta on October 7, and H1 Zeta on October 26, and by H1 Grace in August 2021. Hurricanes contribute importantly to coral reef degradation as they reduce live coral cover and potentially affect structural complexity [37–39]. Therefore, using the data from González-Gómez et al. [35] as a comparative baseline, in 2022, we investigated potential changes in the macrocrustacean communities associated with Limones and Bonanza reefs. As González-Gómez et al. [35] found a higher richness and abundance of macrocrustaceans in the more degraded reef (Bonanza), if habitat degradation increased in these reefs because of the stressors, we would expect richness and/or abundance of macrocrustaceans to increase on these reefs, as also found in several studies, in which diversity measures increased with some level of degradation [27,33,34,36].

## 2. Materials and Methods

### 2.1. Study Area

The Puerto Morelos Reef National Park (PMRNP; Figure 1) is a marine protected area located on the northern coast of the state of Quintana Roo, Mexico, in the Mexican Caribbean. The PMRNP includes a series of reefs that differ in size and structural complexity [7,40], separated from the coast by a shallow (≤5 m) reef lagoon. Two of these reefs, called "Limones" (centered at 20°59.1′ N, 86°47.9′ W) and "Bonanza" (centered at 20°57.6′ N, 86°48.9′ W), are similar in size (~1.5 km in length) but exhibit highly contrasting levels of habitat degradation despite being separated from each other by a distance of ~2 km (Figure 1). Up to 2015, Limones sustained healthy and resilient populations of *Acropora palmata*, with cover values of up to 50% along its central part [7,41]. In contrast, the average live coral cover on Bonanza was ~7%, with extensive areas of relic Acropora skeletons and a predominance of erect macroalgae (>60% cover) [7]. Fishing activities have been banned on both Limones and Bonanza since 1996. However, Bonanza is open to visitation, whereas tourist activities are not permitted on Limones, given the high ecological value of this reef [41].

### 2.2. Macrocrustacean Surveys

For comparative purposes, we used the same methodology as González-Gómez et al. [35]. Sampling was conducted between late October 2021 and July 2022 (hereafter "2022") and consisted of 30 transects (25 m long) laid over the back reef to the crest zones along each reef. The GPS coordinates of the start of each transect laid in 2015 were used as points of origin for the transects laid in 2022. Transects ran parallel to the reef, and the direction of each transect was chosen at random. To minimize potential seasonal effects, sampling was interspersed between reefs throughout the sampling period [42]. Using scuba, all conspicuous decapods (i.e., motile decapods and stomatopods larger than ~1.5 cm [35]) observed within 1 m on both sides of the transect line (i.e., a 50 m² area) were recorded. Individuals were identified in situ with the highest resolution possible, and many were extensively photographed underwater to further help in their identification. Only a few individuals, difficult to identify underwater, were collected in zip-lock bags and taken to the laboratory for their correct identification. Underwater sampling was conducted by two scientific observers with >20 years of experience visually identifying Caribbean macrocrustaceans, and a third observer who was thoroughly trained by repeatedly studying an extensive guide of local crustacean species created in our lab with photos from many different sources, followed by direct identification in the field during several preliminary dives. The two experienced observers were the same in 2015 and 2022. In all cases, the results of the third observer were cross-checked with those of an experienced observer [43,44]. Moreover, we recorded the type of microhabitat (e.g., *Acropora palmata*, *Agaricia* spp., other live corals, *Millepora* spp., dead coral skeletons, coral rubble, gorgonians, algae, anemones, sand, seagrass) in which each macrocrustacean was observed. Field activities were allowed by a permit issued by the Comisión Nacional de Acuacultura y Pesca (PPF/DGOPA-044/20).

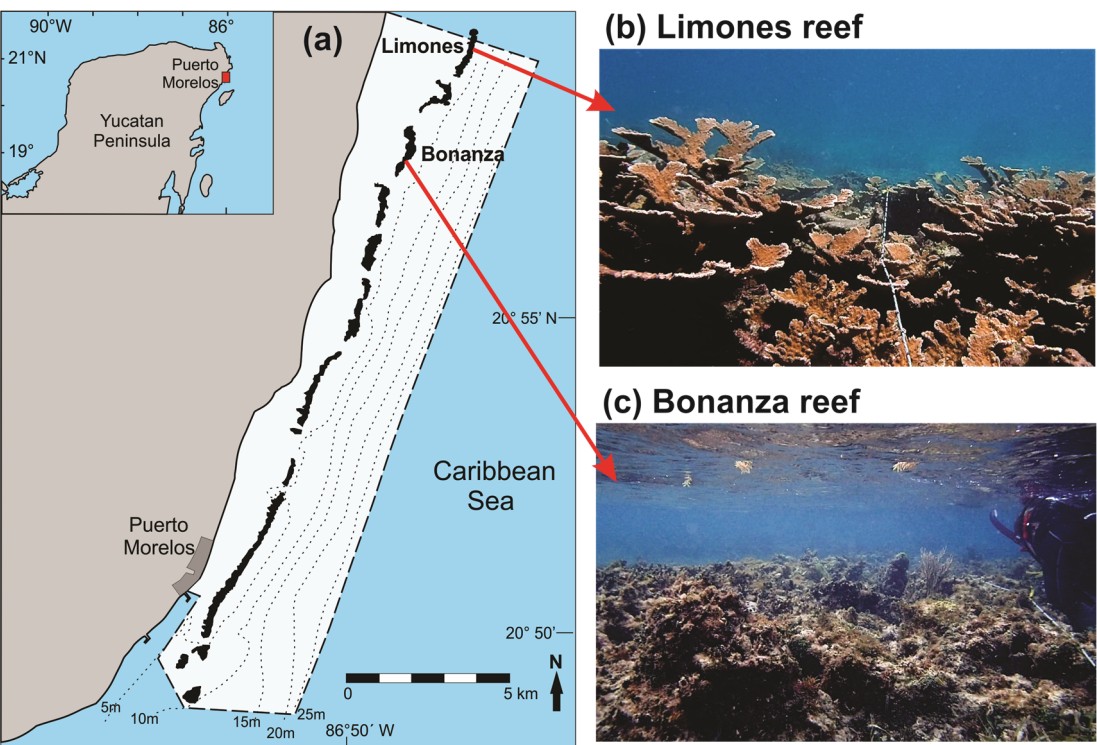

**Figure 1.** (**a**) Schematic map of the Puerto Morelos Reef National Park (PMRNP, dashed line), showing the location of the studied reefs, Limones and Bonanza. The dotted lines denote isobaths. The inset shows the location of PMRNP on the Mexican Caribbean coast. Photographs showing the current state of (**b**) Limones and (**c**) Bonanza (Photo credits: Fernando Negrete-Soto).

### 2.3. Benthic Community Structure

Percent cover of hard coral, different types of macroalgae, and other components of the benthic community are useful in assessing the health of reefs [45]. These percentages of cover were estimated by means of the point intercept method [45]. Data were taken from 16 transects (8 per reef) laid on the central part of each reef in November 2020, shortly after the passage of the three hurricanes, by Estrada-Saldívar et al. [46], and from 23 transects (12 on Limones and 11 on Bonanza) that had been laid in 2015 in the same area of each reef as the 2020 transects [35]. Each transect was marked every 10 cm, thus yielding 100 points per transect. A diver recorded which of the following benthic components was observed under each mark: live hard corals; coralline algae; algal turf; calcareous macroalgae; fleshy macroalgae; cyanobacterial mat; other invertebrates (e.g., zoanthids, *Millepora*, sponges, gorgonians); and other components (e.g., sand, seagrass, hardbottom).

### 2.4. Data Analysis

The percent data on the benthic community structure were logit-transformed [47] and subjected to an exploratory principal component analysis (PCA). Then, a two-factor multivariate analysis of variance (MANOVA) with a general linear model (GLM) approach [48] was used to test for a potential effect of the reef (with two levels, Limones and Bonanza) and year (with two levels, 2015 and 2020) on a combination of the eight groups of benthic components. The transformed data met all MANOVA assumptions. MANOVA results were followed by univariate analyses to examine the individual benthic components.

To examine differences in diversity between reefs and years, we estimated the following ecological indices [49]: species richness, $S$ (number of species); Simpson's dominance, $D = \sum \left(\frac{n_i}{N}\right)^2$, where $n_i$ is the number of individuals of the $i^{\text{th}}$ species, and N is the total number of individuals; Shannon–Wiener's diversity, $H' = -\sum_{i=1}^{S} p_i \log_2 p_i$, where $H'$ is the information contained in the sample (bits/individual), and $p_i = n_i/N$; and Pielou's

evenness, $J' = H'/\log S$. Each index was compared between reefs by sampling year with a Mann–Whitney U test.

We used multivariate techniques with PRIMER 6 v6.1.9 (PRIMER-E Ltd.) to analyze and compare the community composition of macrocrustaceans between reefs and sampling years. To better visualize potential changes, differences in the taxonomic composition between Limones and Bonanza were analyzed separately in 2022 and 2015 by non-metric multidimensional scaling (nMDS) on square-root transformed data using the Bray–Curtis similarity measure [50]. For each sampling year, a one-way analysis of similarity (ANOSIM) was used to test the statistical significance of the observed differences in the macrocrustacean assemblages between reefs. This test provides an R-value indicative of the degree of difference between samples as well as a *p*-value for the significance of that difference. R-values close to 0 indicate little difference, while values close to 1 indicate a large difference in sample composition [51]. We then did a similarity percentage analysis (SIMPER) using the entire dataset to identify the species responsible for the observed differences in community composition between both reefs and years [50].

## 3. Results

### 3.1. Benthic Community Structure

The first two components of the PCA performed with the transformed data on benthic community components explained 63% of the total variance (Figure 2). The first component, which explained 43.1% of the variance, was negatively correlated with live hard coral (loading: −0.694) and positively correlated with fleshy macroalgae (0.493). The second component, which explained 19.9% of the variance, was negatively correlated with other components (−0.709) and positively correlated with coralline algae (0.543) (Figure 2). In both years, transects on Limones differed from those on Bonanza along the first component. However, the smaller convex hulls of both reefs in 2020 suggest a more similar benthic community structure among transects within each reef in 2020 than in 2015.

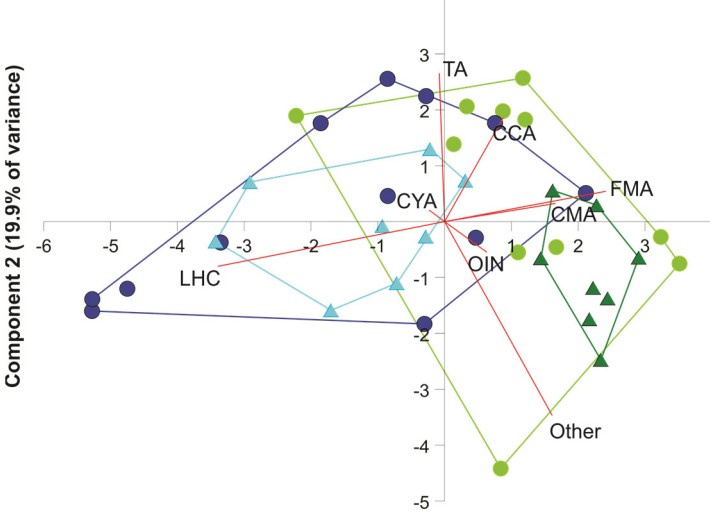

**Figure 2.** Principal Components Analysis (PCA) bi-plot on logit-transformation of percent cover of benthic components over the two studied reefs in 2015 and 2020. Each symbol represents a transect, and the lines are the convex hulls. Dark blue dots and lines: Limones 2015; aqua triangles and lines: Limones 2020; light green dots and lines: Bonanza 2015; dark green triangles and lines: Bonanza 2020. LHC, live hard coral; TA, turf algae; FMA, fleshy macroalgae; CMA, calcareous macroalgae; CCA, coralline algae; CYAN, cyanobacterial mats; OIN, other sessile invertebrates; Other, other components (sand, seagrass, hardbottom).

There was a significant effect of reef (two-factor MANOVA, Wilk's lambda = 0.401, $F_{8,28}$ = 5.224, $p$ < 0.001) and year (Wilk's lambda = 0.421, $F_{8,28}$ = 4.812, $p$ < 0.001) on the combination of benthic components, but not of the reef × year interaction (Wilk's lambda = 0.793, $F_{8,28}$ = 0.913, $p$ = 0.520). Univariate analyses revealed significant differences between reefs in percent cover of live hard coral ($F_{1,35}$ = 28.342, $p$ < 0.001) (higher in Limones), calcareous macroalgae ($F_{1,35}$ = 25.798, $p$ < 0.001) (higher in Bonanza), and fleshy macroalgae ($F_{1,35}$ = 8.389, $p$ = 0.003) (also higher in Bonanza) (Figure 3) and significant differences between years in percent cover of cyanobacterial mats ($F_{1,35}$ = 5.965, $p$ = 0.02) (higher in 2015), other invertebrates ($F_{1,35}$ = 24.190, $p$ < 0.001) (higher in 2020), and other components ($F_{1,35}$ = 4.875, $p$ = 0.034) (also higher in 2020) (Figure 3), but no significant interaction effect for any component. Therefore, both reefs underwent similar changes in the percent cover of benthic components between 2015 and 2020. However, the variability of the main benthic components tended to be lower in 2022 relative to 2015 (Figure 3).

### 3.2. Macrocrustacean Assemblages

Overall, in 2022 we recorded 11,904 individuals of 67 species. This number of individuals contrasts with that recorded in 2015 (4876 individuals of 65 species) [35]. In 2015, more species and individuals were observed on Bonanza (49 and 2805, respectively) than on Limones (39 and 2071, respectively) (Table A1 in Appendix A). In 2022, the number of species became more similar between Bonanza (49) and Limones (50), but the difference in the number of individuals between reefs became more pronounced, with 8244 individuals observed on Bonanza and 3660 on Limones. However, of the total individuals observed on Bonanza, ~5000 were blue-legged hermit crabs *Clibanarius tricolor* that were recorded within one single belt transect (Table A1 in Appendix A).

Macrocrustaceans on both reefs comprised decapods of infraorders Brachyura, Anomura, Caridea, Achelata, Axiidea, Gebiidea, and Stenopodidea, the Superfamily Penaeoidea, as well as stomatopods (Order Stomatopoda). The proportion of species and individuals within each of these taxa by reef and year are shown in Figure 4a–d. In both years, true crabs (Brachyura) made up the majority of species, followed by hermit and porcellanid crabs (Anomura) and caridean shrimps (Caridea) (Figure 4a,c). Spiny and slipper lobsters (Achelata), ghost and mud shrimps (Axiidea and Gebiidea), penaeid shrimps (Penaeoidea), banded shrimps (Stenopodidea), and stomatopods comprised relatively few species, especially in 2022 (Figure 4c), when the percentage of species of Anomura increased on both reefs relative to 2015 (Figure 4a). Regardless, the changes in proportions of species by higher taxon were not significant (3-D contingency table analysis, $\chi^2_{16}$ = 10.418, $p$ = 0.843). In terms of the number of individuals, Brachyura and Anomura were by far the most abundant taxa on both reefs. However, anomurans were significantly more abundant and brachyurans less abundant on Bonanza in 2022 (Figure 4d) than in 2015 (Figure 4c) ($\chi^2_{16}$ = 1750.82, $p$ < 0.0001).

The ecological indices of species richness, diversity, evenness, and dominance, which differed significantly between reefs in 2015 [35], became more similar between reefs by 2022 (Figure 5). This change is not ascribable to Bonanza, as on this reef all indices were similar between 2015 and 2022. In contrast, relative to 2015, species richness, diversity, and evenness increased in Limones in 2022, whereas dominance decreased (Figure 5).

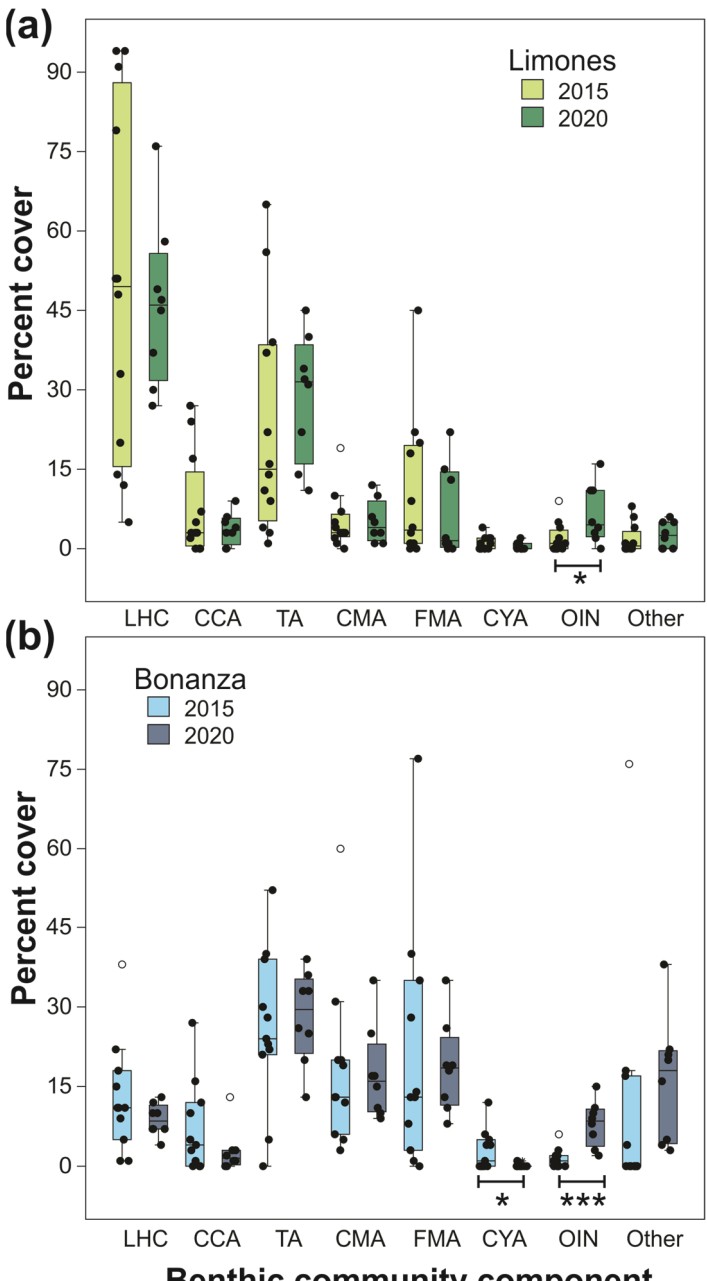

**Figure 3.** Box plots of percent cover of different components of the benthic community estimated in 2015 and 2022 on (**a**) Limones and (**b**) Bonanza. Within each reef, components whose percent cover differed between years are marked. * $p < 0.05$; *** $p < 0.001$. Central lines in boxplots correspond to medians; box extremities indicate interquartile range (IQR, first and third quartiles); whiskers include all data within 1.5 times the IQR; white dots outside the whiskers denote outliers. LHC, live hard coral; CCA, coralline algae; TA, turf algae; CMA, calcareous macroalgae; FMA, fleshy macroalgae; CYA, cyanobacterial mats; OIN, other sessile invertebrates; Other, other components (sand, seagrass, hard bottom).

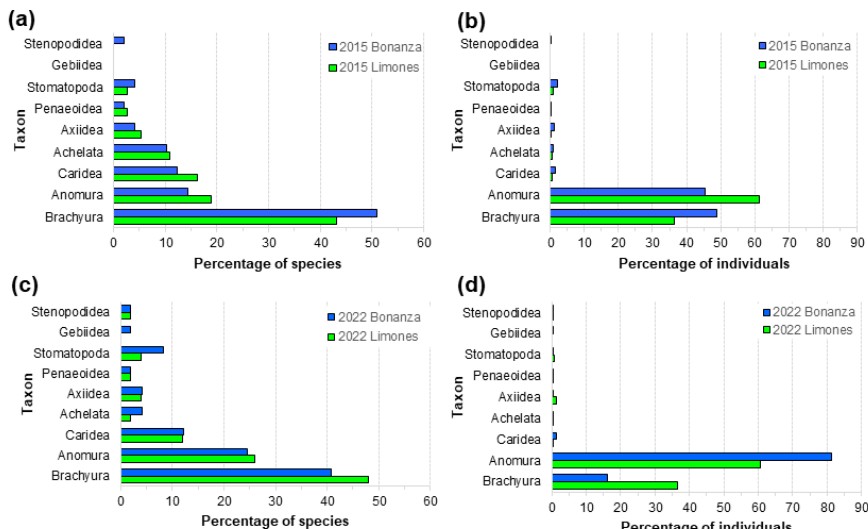

**Figure 4.** Percentage of the total families recorded on Limones reef (green columns) and Bonanza reef (blue columns) (**a**) in 2015 (N = 37 and 48, respectively) and (**c**) in 2022 (N = 50 and 49, respectively), and percentage of the total individuals recorded on Limones and Bonanza reefs (**b**) in 2015 (N = 2071 and 2805, respectively) and (**d**) in 2022 (N = 3660 and 8244, respectively).

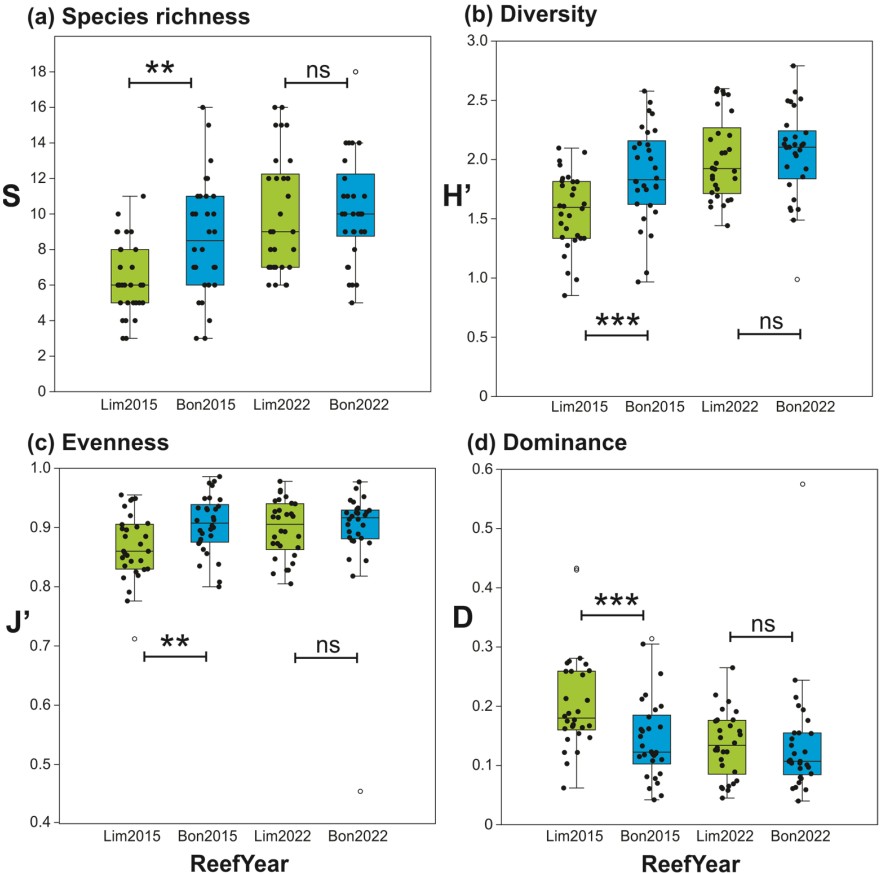

**Figure 5.** Box plots of ecological indices for macrocrustaceans by reef (Limones, green boxes; Bonanza, blue boxes) and year (2015 and 2022), with results of statistical comparison between reefs by year (** $p < 0.01$; *** $p < 0.001$, n.s. = not significant). (**a**) Species richness; (**b**) Shannon–Wiener's diversity; (**c**) Pielou's evenness; (**d**) Simpson's dominance. Central lines in boxplots correspond to medians; box extremities indicate interquartile range (IQR, first and third quartiles); whiskers include all data within 1.5 times the IQR; white dots outside the whiskers denote outliers.

The nMDS 2-D ordination plots for both years showed overlap in the macrocrustacean assemblages between Limones and Bonanza (Figure 6), but to a much greater extent in 2022 (Figure 6b) than in 2015 (Figure 6a). The stress values (0.2) were relatively high, but 3-D ordination plots (not shown) with stress values of 0.14 confirmed the overlap between both reefs, especially in 2022. Indeed, ANOSIM yielded a lower value of R in 2022 (R = 0.136) than in 2015 (R = 0.256). Results of SIMPER (Table A2 in Appendix B) showed substantial similarity in the community composition of crustaceans among transects within each reef, ranging from 46.0% for Bonanza in 2015 to 49.7% on Limones in 2022, with four to eight species accounting for ~90% of similarity per reef each year. In 2015, the three most common species by reef were *Calcinus tibicen*, *Mithraculus coryphe*, and *Domecia acanthophora* on Limones (jointly accounting for 87% of the similarity), and *M. coryphe*, *C. tibicen*, and *Neogonodactylus oerstedii* on Bonanza (accounting for 78.5% of the similarity). In contrast, in 2022, the three most common species were the same on both reefs, *M. coryphe*, *C. tibicen*, and *Pagurus brevidactylus*, jointly accounting for 78% of the similarity on Limones and 69% of the similarity on Bonanza. Importantly, *D. acanthophora*, which was the third most important contributor to the similarity in Limones in 2015, with 8.3%, descended to sixth place in 2022, with only 2.1%. Between reefs, dissimilarity in crustacean composition was higher in 2015 (59%) than in 2022 (55%). The three main contributors to the dissimilarity between reefs were *C. tibicen*, *M. coryphe*, and *D. acantophora* in 2015 and *M. coryphe*, *C. tibicen*, and *P. brevidactylus* in 2022 (Table A2 in Appendix B).

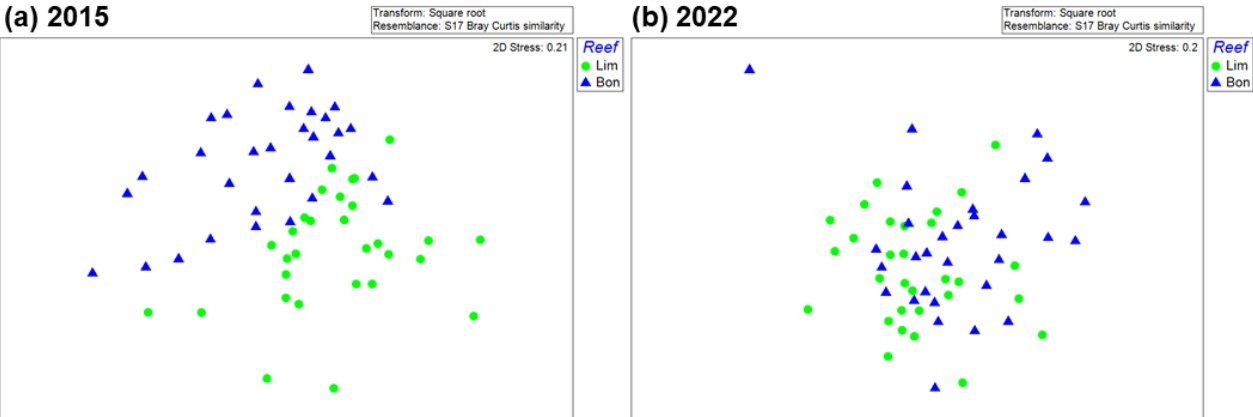

**Figure 6.** Non-metric multidimensional (nMDS) ordination of macrocrustacean community composition on Limones reef (green dots) and Bonanza reef (blue triangles) in (**a**) 2015 and (**b**) 2022, based on species abundances. Each symbol denotes a transect (N = 30 in every case).

*3.3. Microhabitat use by Macrocrustaceans*

The types of microhabitats more commonly occupied by macrocrustaceans on each reef varied with sampling year (Figure 7). The differences were more pronounced in Limones, where the most commonly occupied microhabitats in 2015 were, in descending order, *Millepora* spp., *A. palmata*, coral rubble, and dead coral, whereas in 2022, the most commonly occupied microhabitats were dead coral, rubble, *Millepora* spp., and *Agaricia* spp. Differences in microhabitat use between years on this reef were significant ($\chi^2_9 = 945.92$, $p < 0.0001$), with the most notable differences being the percentages of crustaceans observed on *A. palmata* and *Millepora* spp., 19.6% and 49%, respectively, in 2015, but only 4.1% and 22.3%, respectively, in 2022 (Figure 7). On Bonanza, the greatest percentages of crustaceans were observed in rubble, dead coral, macroalgae, *Agaricia* spp., and *Millepora* spp. in 2015, but in dead coral, rubble, *Agaricia* spp., and macroalgae in 2022. The between-years differences in microhabitat use by macrocrustaceans on Bonanza were also significant ($\chi^2_9 = 621.38$, $p < 0.0001$). Importantly, the percentage of macrocrustaceans using *Millepora* spp. on Bonanza declined from 13.1% in 2015 to 2.9% in 2022 (Figure 7).

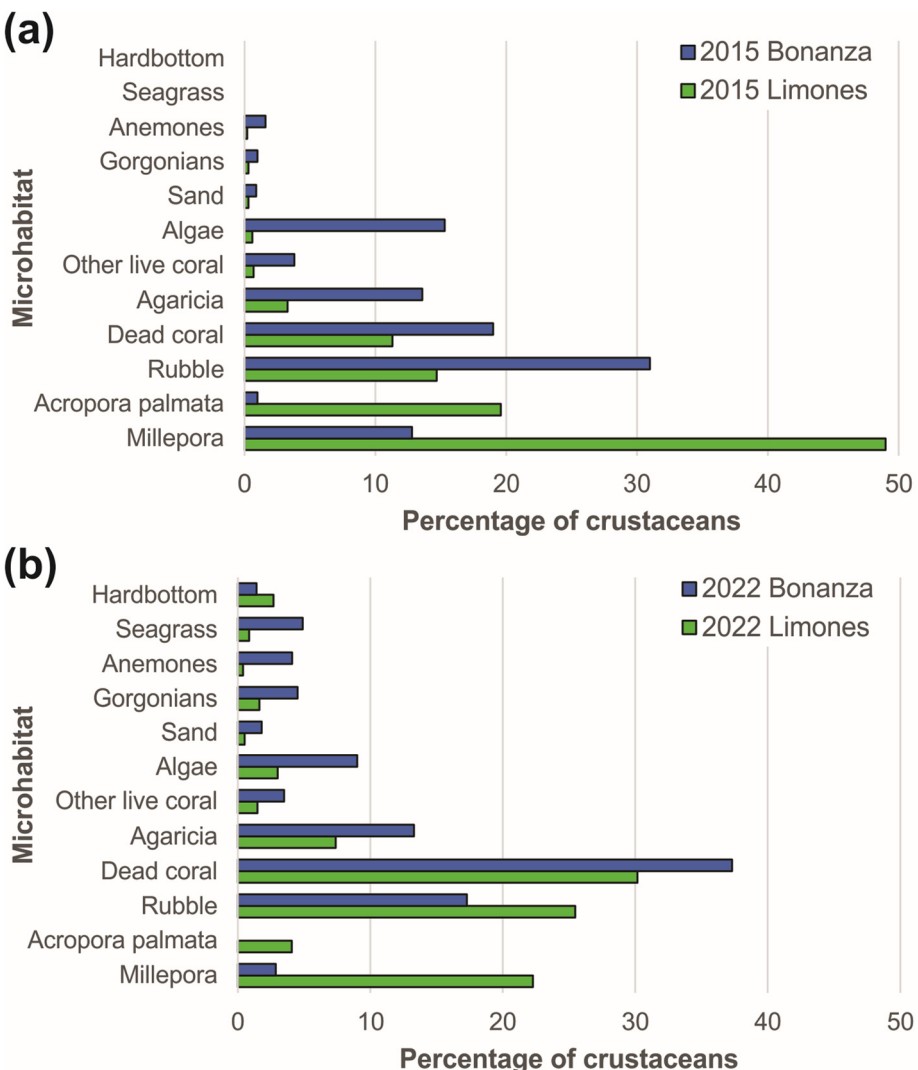

**Figure 7.** Percentages of crustaceans using different types of microhabitats on Limones reef (green columns) and Bonanza reef (blue columns) in (**a**) 2015 and (**b**) 2022.

## 4. Discussion

We investigated the potential effects of unprecedented stressors (the massive influxes of *Sargassum*, the outbreak and propagation of SCTLD, and three consecutive hurricanes) on macrocrustacean communities associated with two coral reefs that previously differed in their level of degradation [7]. Although there are inherent difficulties in sampling motile benthic crustaceans in coral reefs, observation by trained divers is still the most efficient way to find these organisms when they are sufficiently large to be seen [52]. However, even for conspicuous organisms, visual censuses have limitations. There may be variability between observers; some taxa (e.g., Alpheidae, Cryptochiridae) are more cryptic than others or may only be seen at night, when they emerge to forage, and turbidity, waves, and currents may make the divers' observations difficult [43,45,53]. In these circumstances, it is advisable to increase the number of replicates [43]. Coral reef biodiversity survey programs usually consider six to eight transects per site [54,55], whereas reef crustaceans have been generally sampled using five to 24 sampling units [transects, quadrats, or coral colonies] per site (e.g., [26,36,56]). Belt transects, in particular, facilitate gathering the data on species density [53]; therefore, González-Gómez et al. [35] used 30 replicate belt transects per reef in 2015. These authors acknowledged that a more exhaustive sampling would surely increase the number of macrocrustacean species recorded in these reefs. However, for comparative

purposes, we repeated the study in 2022 with the same number of transects per reef, using the same starting points, and surveyed by the same scientific observers as in 2015.

Between 2015 and 2022, the macrocrustacean communities of Limones and Bonanza underwent several changes. In 2015, all ecological indices differed significantly between reefs [35]. By 2022, species richness and diversity increased on Limones, but the dominance decreased. On Bonanza, which was already more degraded than Limones in 2015 [6,30], species richness and diversity did not change much between 2015 and 2022. Consequently, all four ecological indices of macrocrustaceans became more similar between reefs in 2022 than in 2015, which, in conjunction with the results of the nMDS and ANOSIM, suggests that, over this seven-year period, the macrocrustacean communities on Limones have become more similar to those on Bonanza. This would appear to support the notion that the diversity and richness of reef-associated macrocrustaceans are higher on degraded reefs due to the greater availability of certain microhabitats, such as rubble and macroalgae growing on the dead corals [27,30,34,35,57,58], and that the degree of degradation has increased on both reefs, especially on Limones.

Yet, the results on the percent cover of benthic community components are not consistent with an increase in reef degradation between 2015 and shortly after the passage of the three 2020 hurricanes. Although the latter survey was conducted eleven months before the macrocrustacean samplings began in October 2021, changes in the benthic community composition would appear unlikely in such a short period [46]. On Limones, the average percent cover of live hard coral remained high in 2020 (46% vs. 49% in 2015), and the only significant change was a higher percent cover of other invertebrates (i.e., sponges, octocorals, hydrozoans, anemones) in 2020. Similarly, on Bonanza, the only significant changes between 2015 and 2020 were an increase in the percent cover of other invertebrates and a decrease in cover of cyanobacterial mats, which generally occupy a very small proportion of reef space and exhibit a pulsing nature [59]. In addition, the effect of SCTLD has been low on Bonanza and negligible on Limones [46] because *A. palmata* is not susceptible to this disease [14]. In both reefs, however, the range in percent cover values decreased for the most abundant benthic components, and the PCA revealed that the benthic community composition became more similar between transects within reefs, potentially reducing the habitat heterogeneity. In other Caribbean coral reefs, individual benthic components, rather than reef complexity, have been shown to have considerable importance to different species of motile invertebrates and fish [60]. Therefore, our results might reflect subtle ecological changes in the integrity of Limones and Bonanza that are not necessarily captured by standard metrics of reef condition, such as coral or macroalgal cover, suggesting that future studies addressing changes in diversity and abundance of reef-associated macrocrustaceans should evaluate the full spectrum of microhabitats used by these animals. For example, the crustacean communities might be susceptible to changes in the amount of sediments and detritus, or an increased number of coral fragments and rubble [46,57,58].

Indeed, the 2020 hurricanes substantially increased the number of coral fragments and rubble on both reefs, but mainly on Limones [46]. Hurricanes contribute to coral reef degradation by breaking branching and foliose coral species [37–39], with the magnitude of the immediate loss increasing both with hurricane intensity and with the time elapsed since the last impact [37,61]. The intensity of the hurricanes that impacted our study area in October 2020 (TS, H2, and H1) and in August 2021 (H1) was relatively low, but the short period between their respective impacts, especially in 2020, resulted in substantial damage to the PMRNP reefs [46]. Branching and foliose corals, such as *A. palmata*, *Agaricia* spp., and *Millepora* spp., were the most damaged. Therefore, Limones, where these species were previously abundant, was among the most highly affected reefs in the area, with almost 10% of coral colonies broken, an additional 8% overturned, and an average of 5 coral fragments per 10 m$^2$ of reef area [46]. Although the breaking of *A. palmata* colonies during a hurricane is not necessarily damaging, as this species mainly reproduces through fragmentation [62], there can be delayed mortality of coral fragments [63]. However, our Limones transects were distributed along the entire reef, whereas the most prominent and healthy stands of *A.*

*palmata* on Limones are located around the central part of the reef and closer to the crest [7]. In this part, the results of the benthic community composition show that the loss of live coral was not so evident. On Bonanza, where *A. palmata* has been far less abundant for years [41], and the level of degradation was already high in 2015 [7,35]; loss of live coral cover in 2022 has been rather attributed to the SCTLD, although hurricanes did damage the previously abundant *Millepora* spp. colonies [46]. Overall, Bonanza was far less damaged than Limones, which is consistent with the notion that degraded coral communities are more resistant to severe storms than healthier coral communities [64].

In addition to the hurricane effects, the PMRNP coral reefs have been subjected to a continuous increase in eutrophication and sedimentation in the reef lagoon [11,65] due to the combined effect of the expansive tourism and urban development in Puerto Morelos and the massive influxes of *Sargassum*. Currently, the dissolved nitrogen concentration in the water and the organic matter in the sediments have reached levels that are considered a threat to the maintenance of the oligotrophic nature of coral reefs [11], and the highest accumulation rates of sediments in the last decades have been measured after the arrival of massive *Sargassum* influxes [65]. These changes, in conjunction with the differential effects of the hurricanes and the SCTLD on the benthic community components of Limones and Bonanza, may underlie some of the changes in macrocrustacean communities.

For example, in both 2015 and 2022, *M. coryphe* and *C. tibicen* were the most abundant species on Bonanza (excluding the transect with ~5000 *C. tricolor*, which will be further addressed). *Mitrhaculus coryphe* is a generalist herbivore that feeds on many types of macroalgae, including species that grow on dead coral and species that are noxious to other herbivores, such as fish [66–69]. On Limones, the abundance of this mithracid crab increased almost four times between 2015 (290 individuals, [35]) and 2022 (1064, this study), probably due to the increase in rubble, which is an important microhabitat for these crabs [35], and the fragments of corals broken by the hurricanes. In Brazil, the abundance of mithracid crabs was greater on reef sites where eutrophication increased habitat degradation and macroalgal abundance [56]. The hermit crab *C. tibicen* was the most abundant species on Limones and the second most abundant on Bonanza (again, excluding *C. tricolor*) for both sampling years, although its abundance decreased in Bonanza and increased in Limones between 2015 and 2022. *Calcinus tibicen* is an omnivorous detritivore [70,71] and has a facultative symbiosis with fire corals (*Millepora* spp.), finding refuge in its branches [72]. Limones still has considerable amounts of *Millepora*, and we observed individuals of *C. tibicen* on most *Millepora* colonies, although many other individuals were found on dead coral, rubble, and other live corals. It is possible that the increase in coral fragments resulting from the hurricanes [46] increased the availability of microhabitats for these hermit crabs in Limones.

One of the most remarkable differences in species abundance between 2015 and 2022 was that of *D. acanthophora* on Limones. Although this small commensal crab has been recorded on eight species of hard corals, several species of *Millepora*, and even sponges [73], it has an important relationship with *A. palmata* [74] and, to a lesser extent, with *Millepora* [35,75]. In 2015, *D. acanthophora* was the second most abundant species on Limones, with 377 individuals [35], but dropped to fifth place in 2022 with only 75 individuals. González-Gómez et al. [35] predicted that the abundance of this crab would decrease if the degradation of Limones increased. Because of the decline in its abundance, *D. acanthophora* was partially responsible for the dissimilitude in the macrocrustacean assemblage of Limones between 2015 and 2022. The decline of *D. acanthophora* may have been caused by the destruction of their main habitat, *A. palmata*, by the hurricanes or by stress. On Pacific reefs, stressed corals and their symbiotic crabs (*Trapezia* spp.) underwent drastic reductions in lipid content that resulted in the death of many individuals [76]. Hurricanes may have differential effects on the motile reef epifauna, depending on the size of the animals and the level of exposure of their particular habitats. For example, immediately after the passage of two hurricanes close to the island of St. Croix, USVI, most taxa were virtually wiped from intertidal zones, but on back-reef habitats similar to the ones we sampled, the abundance

of crabs and some caridean shrimps first declined and then recovered over the following 18 months, whereas other taxa showed the opposite pattern [77]. Whether the abundance of *D. acanthophora* in Limones will return to previous levels remains to be determined. On this reef, the percentages of crustaceans using *A. palmata* and *Millepora* as microhabitats decreased in 2022 relative to 2015. Although the percentage of crustaceans using *Millepora* in 2022 decreased to about one-half of that observed in 2015, it remained relatively high, but, as noted before, *D. acanthophora* appears to prefer *A. palmata* to *Millepora* [35,74]. These results suggest that *D. acanthophora* may be an indicator of the state of conservation of acroporid reefs throughout the Caribbean.

The occurrence of aggregations amounting to ~5000 individuals of *C. tricolor* along a single transect on Bonanza in 2022 is an interesting phenomenon, especially since this species was not reported in 2015 on either Bonanza or Limones [30]. We have no explanation for this phenomenon, but although hermit crabs are omnivorous and even scavengers, they are predominantly detritivores [70,71], so it could be speculated that the aggregations of *C. tricolor* may have been favored by a local increase in food (e.g., organic detritus in sediments [65]). According to Hazlett [70], this diogenid forms stable groups that disperse at night to forage and regroup during the day, when most interactions, such as reproduction or fights for shells, occur. However, other than the extreme abundance of this hermit crab along that one transect, it was only recorded, at much lower abundances, on three additional transects on Bonanza (6–74 individuals) and two transects on Limones (15 and 32 individuals). Therefore, *C. tricolor* was not as frequently recorded as other abundant species were, such as *M. coryphe* or *C. tibicen*, which occurred in all transects on both reefs or as *P. brevidactylus*, which occurred in most transects on both reefs. Hence, it was not an important contributor to the dissimilarity between reefs in 2022. Interestingly, Hazlett [70] remarked that even though *C. tricolor* was the most abundant hermit crab in his samples in Curaçao; it had not been registered in previous studies in that location.

On coral reefs, motile epifauna, of which crustaceans are a substantial component, can use many different microhabitats other than live coral, such as macroalgae, rubble, and holes and crevices in dead coral [57,58,78]. These microhabitats may become increasingly important in supporting coral reef biodiversity and food webs on degraded reefs [26]. The increase in abundance and, to a lesser extent, in the diversity of motile invertebrates on degraded coral reefs, as long as the reef structure is maintained, has been reported by several authors (e.g., [27,30,34–36]) and is considered important for the trophic webs of degraded reefs, as many reef fishes are primarily invertivores, most of them feeding on crustaceans [79,80]. A size-based ecosystem model predicted that as a coral reef system changed from coral to algal turf but maintained reef structure, invertebrates dominated and fish productivity increased by around 23%, but as loss of reef structure continued, all faunal components decreased and productivity dropped by an additional 54% [81]. Therefore, if degraded reefs continue to erode over time, as current trends suggest [3,82,83], the structures that serve as microhabitats for macrocrustaceans may be eliminated. On the other hand, reef fishes (e.g., some Labridae, Haemulidae, Muraenidae, Holocentridae, Serranidae, Lutjanidae, Balistidae) that prey on macrocrustaceans (i.e., crustaceans >3 mm [24]) mostly consume brachyuran crabs [80,84–86]. In contrast, Caribbean hermit crabs sustain lower levels of predation by fishes, but higher levels of predation by other crustaceans (e.g., stomatopods), than their tropical Pacific counterparts [87]. Therefore, decreases in the relative abundance of brachyuran crabs and increases in the relative abundance of hermit crabs, as occurred on Bonanza between 2015 and 2022, may have consequences for the local trophic webs [88].

The effects of multiple drivers and their combined impacts on coral reef communities are very difficult to assess [2,8]. Even though the separate contribution of stressors acting between 2015 and 2022 (*Sargassum* brown tides, SCTLS, hurricane impact) to the observed changes cannot be disentangled, our results suggest that the macrocrustacean communities associated with these reefs have become more similar over time, i.e., exhibit taxonomic homogenization [89]. A similar situation is being recorded for crustacean communities

associated with seagrasses in the Puerto Morelos reef lagoon (P. Briones-Fourzán, unpublished data). Along the reefs of the PMRNP, changes in coral communities over the last three decades have resulted in the structural and functional convergence of the back-reef and the fore-reef zones, with a dominance of low-relief species [20]. Other studies have documented a tendency to homogenize or converge in communities of other taxa on sites located close to populated coastlines or subjected to substantial disturbance events. This has been found in fish assemblages throughout the Caribbean Sea [90], in Florida [91], and elsewhere [92] and has been ascribed to alterations of the structure of ecological communities due to human activities favoring generalists over specialist species [92,93]. Indeed, the increase in macrocrustacean species richness and/or abundance in degraded reefs is not necessarily good news, as the cost of this is usually the loss of habitat specialists and symbiotic species [26,34,89–93]. Whether the taxonomic homogenization of macrocrustaceans on Limones and Bonanza will be followed by a more concerning functional homogenization [89] requires further study.

## 5. Conclusions

The macrocrustacean communities associated with the Limones and Bonanza reefs have undergone changes between 2015 and 2022. The ecological indices of species richness, diversity, evenness, and dominance, which in 2015 differed significantly between both reefs, became similar in 2022, and the greater overlap in the crustacean communities in 2022 compared to 2015 suggests a taxonomic homogenization. Despite the gradual increase in eutrophication, sedimentation, and pollution of the Puerto Morelos reef lagoon over time and the series of events (hurricanes, disease outbreak, and sargassum influxes) that impacted the Puerto Morelos reefs between 2015 and 2022, reef degradation did not significantly increase during this seven-year period. Therefore, our results might reflect subtle ecological changes in the integrity of Limones and Bonanza (e.g., changes in the amount of sediments, detritus, coral fragments, and/or rubble) that are not necessarily captured by standard methods of evaluating reef condition, but to which the crustacean communities might be susceptible. That is, the full spectrum of microhabitats used by macrocrustaceans should be evaluated.

**Supplementary Materials:** The following supporting information can be downloaded at: https://www.mdpi.com/article/10.3390/d15070809/s1, Supplementary Information: crustacean dataset, ecological indices, benthic components.

**Author Contributions:** Conceptualization, M.K.D., P.B.-F. and E.L.-Á.; data curation, M.K.D., P.B.-F. and C.B.-O.; formal analysis, M.K.D. and P.B.-F.; funding acquisition, P.B.-F.; investigation, M.K.D., P.B.-F., C.B.-O., F.N.-S., L.Á.-F. and E.L.-Á.; methodology, M.K.D., P.B.-F., C.B.-O., F.N.-S., L.Á.-F. and E.L.-Á.; project administration, P.B.-F. and F.N.-S.; resources, P.B.-F., F.N.-S and E.L.-Á.; supervision, P.B.-F., L.Á.-F. and E.L.-Á.; validation, M.K.D., P.B.-F. and C.B.-O.; visualization, M.K.D., P.B.-F. and E.L.-Á.; writing—original draft, P.B.-F.; writing—review and editing, M.K.D., L.Á.-F., P.B.-F., C.B.-O., F.N.-S. and E.L.-Á. All authors have read and agreed to the published version of the manuscript.

**Funding:** This research and the APC were funded by National Autonomous University of Mexico: DGAPA-PAPIIT- IN205921, awarded to P.B.-F. M.K.D. received an MSc scholarship from Consejo Nacional de Ciencia y Tecnología (Mexico).

**Institutional Review Board Statement:** Not applicable.

**Data Availability Statement:** The data presented in this study are available in the Supplementary Materials (Excell file Supplementary Information).

**Acknowledgments:** We thank Esmeralda Pérez-Cervantes and Nuria Estrada-Saldívar for their support in the collection of the data on benthic components. We also appreciate the technical support of Edgar Escalante-Mancera and Miguel Gómez-Reali.

**Conflicts of Interest:** The authors declare no conflict of interest. The funders had no role in the design of the study, in the collection, analyses, or interpretation of data, in the writing of the manuscript, or in the decision to publish the results.

# Appendix A

**Table A1.** Crustacean species (in alphabetical order within higher taxa) and number of individuals registered by reef (Limones, Bonanza) in two sampling years (2015 and 2022), Puerto Morelos Reef National Park, Mexico (visual censuses, N = 30 belt transects, 25 m × 2 m, per reef and sampling year).

| | Limones | | Bonanza | |
|---|---|---|---|---|
| **Species** | **2015** | **2022** | **2015** | **2022** |
| ORDER DECAPODA | | | | |
| Superfamily Penaeoidea | | | | |
| *Metapenaeopsis goodei* (Smith, 1885) | 1 | 1 | 1 | 1 |
| Infraorder Achelata | | | | |
| *Panulirus argus* (Latreille, 1804) | 4 | 5 | 18 | 6 |
| *Panulirus guttatus* (Laterille, 1804) | 3 | 0 | 1 | 0 |
| *Parribacus antarcticus* (Lund, 1793) | 0 | 0 | 1 | 0 |
| *Phyllamphion gundlachi* (von Martens, 1878) | 1 | 0 | 0 | 0 |
| *Scyllarides aequinoctialis* (Lund, 1793) | 6 | 0 | 0 | 6 |
| Infraorder Anomura | | | | |
| Anomuran A | 0 | 0 | 1 | 0 |
| *Calcinus tibicen* (Herbst, 1791) | 1143 | 1462 | 1002 | 869 |
| *Clibanarius tricolor* (Gibbes, 1850) | 0 | 47 | 0 | 5094 |
| *Pachycheles pilosus* (H. Milne Edwards, 1837) | 3 | 4 | 0 | 0 |
| *Paguristes anomalus* (Bouvier, 1918) | 15 | 126 | 66 | 205 |
| *Paguristes cadenati* (Forest, 1954) | 18 | 59 | 0 | 0 |
| *Paguristes erythrops* (Holthuis, 1959) | 18 | 59 | 0 | 0 |
| *Paguristes puncticeps* (Benedict, 1901) | 4 | 57 | 19 | 53 |
| *Paguristes tortugae* (Schmitt, 1933) | 0 | 74 | 84 | 40 |
| *Pagurus brevidactylus* (Stimpson, 1859) | 48 | 348 | 97 | 381 |
| *Pagurus marshi* (Benedict, 1901) | 0 | 0 | 0 | 42 |
| *Pagurus provenzanoi* (Forest and de Saint Laurent, 1968) | 0 | 11 | 0 | 7 |
| *Petrolisthes caribensis* (Werding, 1983) | 0 | 0 | 0 | 1 |
| *Petrolisthes galathinus* (Bosc, 1801) | 38 | 11 | 5 | 10 |
| *Phimochirus holthuisi* (Provenzano, 1961) | 5 | 1 | 0 | 0 |
| Porcellanid A | 0 | 3 | 0 | 0 |
| Porcellanid B | 0 | 6 | 0 | 0 |
| *Pylopaguridium markhami* (McLaughlin anf Lemaitre, 2001) | 0 | 15 | 0 | 0 |
| Infraorder Axiidae | | | | |
| *Axiopsis serratifrons* (A. Milne-Edwards, 1873) | 4 | 44 | 12 | 17 |
| *Corallianassa longiventris* (A. Milne-Edwards, 1870) | 1 | 10 | 19 | 28 |
| Infraorder Brachyura | | | | |
| *Achelous sebae* (H. Milne-Edwards, 1834) | 1 | 1 | 0 | 1 |
| *Actaea acantha* (H. Milne-Edwards, 1834) | 0 | 0 | 2 | 1 |
| *Amphithrax aculeatus* (Herbst, 1790) | 11 | 23 | 45 | 7 |
| *Calappa gallus* (Herbst, 1803) | 0 | 1 | 0 | 1 |
| *Carpilius corallinus* (Herbst, 1783) | 0 | 0 | 1 | 0 |
| *Domecia acanthophora* (Desbonne in Desbonne and Schramm, 1867) | 377 | 75 | 45 | 0 |
| *Epialtus bituberculatus* (H. Milne Edwards, 1834) | 0 | 0 | 0 | 5 |
| *Epialtus longirostris* (Stimpson, 1860) | 0 | 2 | 0 | 0 |
| Grapsoid A | 0 | 0 | 0 | 1 |
| *Macrocoeloma diplacanthum* (Stimpson, 1860) | 0 | 1 | 5 | 28 |
| *Macrocoeloma subparellelum* (Stimpson, 1860) | 0 | 2 | 14 | 12 |
| *Macrocoeloma trispinosum* (Latreille, 1825) | 0 | 1 | 2 | 0 |
| Majoid A | 2 | 0 | 0 | 0 |
| Majoid B | 0 | 0 | 1 | 0 |
| Majoid C | 0 | 0 | 1 | 0 |
| Majoid D | 0 | 0 | 1 | 0 |

**Table A1.** *Cont.*

| Species | Limones | | Bonanza | |
|---|---|---|---|---|
| | **2015** | **2022** | **2015** | **2022** |
| *Maguimithrax spinosissimus* (Lamarck, 1818) | 1 | 0 | 0 | 0 |
| *Mithraculus cinctimanus* (Stimpson, 1860) | 3 | 0 | 3 | 1 |
| *Mithraculus coryphe* (Herbst, 1801) | 290 | 1064 | 1071 | 1021 |
| *Mithraculus forceps* (A. Milne-Edwards, 1875) | 0 | 8 | 2 | 9 |
| *Mithraculus sculptus* (Lamarck, 1818) | 17 | 53 | 70 | 168 |
| *Mithrax hispidus* (Herbst, 1790) | 1 | 9 | 0 | 0 |
| *Mithrax pleuracanthus* (Stimpson, 1871) | 0 | 3 | 0 | 0 |
| *Nemausa acuticornis* (Stimpson, 1871) | 2 | 7 | 0 | 1 |
| *Nonala holderi* (Stimpson, 1871) | 0 | 0 | 7 | 0 |
| *Omalacantha bicornuta* (Latreille, 1825) | 1 | 9 | 43 | 25 |
| *Percnon gibbesi* (H. Milne-Edwards, 1853) | 7 | 11 | 8 | 2 |
| *Pitho lherminieri* (Desbonne in Desbonne and Schramm, 1867) | 0 | 11 | 1 | 25 |
| *Pitho mirabilis* (Herbst, 1794) | 0 | 0 | 1 | 0 |
| *Podochela macrodera* (Stimpson, 1860) | 0 | 2 | 1 | 0 |
| *Ratha longimana* (H. Milne-Edwards, 1834) | 0 | 17 | 2 | 0 |
| *Stenorhynchus seticornis* (Herbst, 1788) | 0 | 0 | 1 | 2 |
| *Teleophrys ruber* (Stimpson, 1871) | 40 | 21 | 95 | 5 |
| *Thoe puella* (Stimpson, 1860) | 0 | 0 | 0 | 2 |
| *Williamstimpsonia denticulatus* (White, 1848) | 0 | 8 | 0 | 11 |
| Xanthoid A | 1 | 1 | 0 | 0 |
| Xanthoid B | 1 | 1 | 0 | 0 |
| Xanthoid C | 0 | 0 | 1 | 0 |
| Xanthoid D | 0 | 0 | 1 | 0 |
| Xanthoid E | 0 | 0 | 1 | 0 |
| Infraorder Caridea | | | | |
| *Alpheus armatus* (Rathbun, 1901) | 4 | 4 | 19 | 6 |
| *Ancylomenes pedersoni* (Chace, 1958) | 0 | 0 | 2 | 6 |
| *Brachycarpus biunguiculatus* (Lucas, 1846) | 0 | 0 | 1 | 0 |
| Caridean A | 3 | 2 | 0 | 0 |
| Caridean B | 0 | 1 | 0 | 0 |
| *Cinetorhynchus manningi* (Okuno, 1996) | 2 | 1 | 0 | 0 |
| *Cinetorhynchus rigens* (Gordon, 1936) | 1 | 0 | 1 | 0 |
| *Lysmata wurdemanni* (Gibbes, 1850) | 1 | 0 | 1 | 0 |
| *Periclimenes rathbunae* Schmitt, 1924 | 0 | 5 | 0 | 3 |
| *Periclimenes yucatanicus* (Ives, 1891) | 0 | 0 | 0 | 2 |
| *Synalpheus* sp. | 1 | 0 | 0 | 0 |
| *Thor dicaprio* (Anker and Baeza, 2021) | 0 | 4 | 19 | 89 |
| *Trachycaris rugosa* (Spence Bate, 1888) | 0 | 0 | 0 | 1 |
| Infraorder Gebiidea | | | | |
| *Thalassina* sp. | 0 | 0 | 0 | 2 |
| Infraorder Stenopodidea | | | | |
| *Stenopus hispidus* (Olivier, 1811) | 0 | 2 | 3 | 2 |
| ORDER STOMATOPODA | | | | |
| *Neogonodactylus bredini* (Manning, 1969) | 0 | 0 | 0 | 1 |
| *Neogonodactylus oerstedii* (Hansen, 1895) | 15 | 24 | 57 | 38 |
| *Neogonodactylus torus* (Manning, 1969) | 0 | 2 | 1 | 8 |
| *Pseudosquilla ciliata* (Fabricius, 1787) | 0 | 0 | 0 | 1 |
| Total individuals | 2071 | 3660 | 2805 | 8244 |
| Total species | 37 | 50 | 49 | 49 |

## Appendix B

**Table A2.** Similarity measures within and between reefs (Limones, Bonanza) and years (2015, 2022). Analysis of similarity percentage (SIMPER) for macrocrustacean communities within Limones reef and Bonanza reef in 2015 and 2022, and of dissimilarity percentage between reefs and years. Av. Abund: average abundance; Av.Sim: average similarity; Sim/SD: similarity/standard deviation; Contrib%: contribution in %; Cum%: cumulative contribution in %; Av.Diss: average dissimilarity; Diss/SD, dissimilarity/standard deviation. Cum% up to 90% is included in each group, Cum.% does not reach 100% in order to facilitate interpretation.

**(A) Limones 2015**
Average similarity: 48.33

| Species | Av.Abund | Av.Sim | Sim/SD | Contrib% | Cum% |
|---|---|---|---|---|---|
| *Calcinus tibicen* | 5.71 | 27.34 | 2.74 | 56.57 | 56.57 |
| *Mithraculus coryphe* | 2.66 | 10.75 | 1.66 | 22.25 | 78.82 |
| *Domecia acanthophora* | 2.22 | 4.00 | 0.50 | 8.27 | 87.09 |
| *Petrolisthes galathinus* | 0.73 | 1.48 | 0.48 | 3.07 | 90.16 |

**(B) Limones 2022**
Average similarity: 50.06

| Species | Av.Abund | Av.Sim | Sim/SD | Contrib% | Cum% |
|---|---|---|---|---|---|
| *Mithraculus coryphe* | 5.43 | 15.37 | 2.7 | 30.71 | 30.71 |
| *Calcinus tibicen* | 6.07 | 15.13 | 1.55 | 30.22 | 60.93 |
| *Pagurus brevidactylus* | 3.08 | 8.61 | 1.69 | 17.2 | 78.12 |
| *Paguristes puncticeps* | 1.09 | 2.33 | 0.85 | 4.66 | 82.78 |
| *Paguristes anomalus* | 1.43 | 2.28 | 0.57 | 4.56 | 87.34 |
| *Domecia acanthophora* | 0.9 | 1.05 | 0.4 | 2.09 | 89.43 |
| *Axiopsis serratifrons* | 0.76 | 0.98 | 0.53 | 1.95 | 91.38 |

**(C) Bonanza 2015**
Average similarity: 46.01

| Species | Av.Abund | Av.Sim | Sim/SD | Contrib% | Cum% |
|---|---|---|---|---|---|
| *Mithraculus coryphe* | 5.21 | 17.79 | 3.02 | 38.66 | 38.66 |
| *Calcinus tibicen* | 5.13 | 15.53 | 1.88 | 33.75 | 72.41 |
| *Neogonodactylus oerstedii* | 1.08 | 2.81 | 0.85 | 6.10 | 78.52 |
| *Pagurus brevidactylus* | 1.32 | 2.50 | 0.74 | 5.43 | 83.95 |
| *Mithraculus sculptus* | 1.07 | 2.16 | 0.60 | 4.70 | 88.64 |
| *Paguristes tortugae* | 0.93 | 0.83 | 0.35 | 1.81 | 90.45 |

**(D) Bonanza 2022**
Average similarity: 46.23

| Species | Av.Abund | Av.Sim | Sim/SD | Contrib% | Cum% |
|---|---|---|---|---|---|
| *Mithraculus coryphe* | 5.56 | 16.88 | 2.65 | 36.51 | 36.51 |
| *Calcinus tibicen* | 4.26 | 8.16 | 1.16 | 17.65 | 54.17 |
| *Pagurus brevidactylus* | 2.99 | 6.83 | 1.22 | 14.77 | 68.93 |
| *Paguristes anomalus* | 1.96 | 3.34 | 0.74 | 7.23 | 76.16 |
| *Mithraculus sculptus* | 1.71 | 3.13 | 0.79 | 6.78 | 82.94 |
| *Thor dicaprio* | 1.09 | 1.37 | 0.51 | 2.95 | 85.89 |
| *Paguristes puncticeps* | 0.86 | 1.12 | 0.47 | 2.42 | 88.32 |
| *Neogonodactylus oerstedii* | 0.75 | 1.05 | 0.52 | 2.27 | 90.59 |

**Table A2.** *Cont.*

(E) Limones 2015 vs. Limones 2022
Average dissimilarity = 59.69

| | Lim2015 | Lim2022 | | | | |
|---|---|---|---|---|---|---|
| Species | Av.Abund. | Av.Abund. | Av.Dissim. | Dis/SD | Contrib% | Cum% |
| *Calcinus tibicen* | 5.71 | 6.07 | 8.07 | 1.37 | 13.52 | 13.52 |
| *Mithraculus coryphe* | 2.66 | 5.43 | 7.79 | 1.31 | 13.05 | 26.57 |
| *Pagurus brevidactylus* | 0.70 | 3.08 | 6.37 | 1.61 | 10.68 | 37.25 |
| *Domecia acanthophora* | 2.22 | 0.90 | 5.44 | 0.94 | 9.11 | 46.36 |
| *Paguristes anomalus* | 0.20 | 1.43 | 3.50 | 0.96 | 5.86 | 52.22 |

(F) Bonanza 2015 vs. Bonanza 2022
Average dissimilarity = 57.64

| | Bon2015 | Bon2022 | | | | |
|---|---|---|---|---|---|---|
| Species | Av.Abund. | Av.Abund. | Av.Dissim. | Dis/SD | Contrib% | Cum% |
| *Calcinus tibicen* | 5.13 | 4.26 | 7.39 | 1.30 | 12.82 | 12.82 |
| *Mithraculus coryphe* | 5.21 | 5.56 | 5.19 | 1.15 | 9.01 | 21.83 |
| *Pagurus brevidactylus* | 1.32 | 2.99 | 4.82 | 1.31 | 8.37 | 30.20 |
| *Paguristes anomalus* | 0.70 | 1.96 | 3.98 | 1.12 | 6.90 | 37.09 |
| *Mithraculus sculptus* | 1.07 | 1.71 | 3.35 | 1.04 | 5.82 | 42.91 |
| *Clibanarius tricolor* | 0.00 | 2.85 | 2.90 | 0.26 | 5.03 | 47.94 |
| *Thor dicaprio* | 0.31 | 1.09 | 2.39 | 0.88 | 4.15 | 52.09 |

(G) Limones 2015 vs. Bonanza 2015
Average dissimilarity = 58.97

| | Lim2015 | Bon2015 | | | | |
|---|---|---|---|---|---|---|
| Species | Av.Abund. | Av.Abund. | Av.Dissim. | Dis/SD | Contrib% | Cum% |
| *Calcinus tibicen* | 5.71 | 5.13 | 8.61 | 1.21 | 14.61 | 14.61 |
| *Mithraculus coryphe* | 2.66 | 5.21 | 8.13 | 1.34 | 13.79 | 28.39 |
| *Domecia acanthophora* | 2.22 | 0.46 | 6.05 | 0.80 | 10.26 | 38.65 |
| *Mithraculus sculptus* | 0.70 | 1.32 | 3.48 | 1.15 | 5.90 | 44.55 |
| *Pagurus brevidactylus* | 0.40 | 1.07 | 3.00 | 0.96 | 5.09 | 49.64 |
| *Teleophrys ruber* | 0.71 | 0.88 | 2.91 | 0.92 | 4.93 | 54.57 |

(H) Limones 2022 vs. Bonanza 2022
Average dissimilarity = 54.85

| | Lim2022 | Bon2022 | | | | |
|---|---|---|---|---|---|---|
| Species | Av.Abund. | Av.Abund. | Av.Dissim. | Dis/SD | Contrib% | Cum% |
| *Calcinus tibicen* | 6.07 | 4.26 | 7.8 | 1.37 | 14.23 | 14.23 |
| *Mithraculus coryphe* | 5.43 | 5.56 | 4.56 | 1.28 | 8.31 | 22.54 |
| *Pagurus brevidactylus* | 3.08 | 2.99 | 3.58 | 1.19 | 6.53 | 29.07 |
| *Paguristes anomalus* | 1.43 | 1.96 | 3.43 | 1.21 | 6.26 | 35.32 |
| *Clibanarius tricolor* | 0.32 | 2.85 | 3.15 | 0.29 | 5.74 | 41.06 |
| *Mithraculus sculptus* | 0.83 | 1.71 | 3.05 | 1.03 | 5.56 | 46.63 |
| *Paguristes tortugae* | 0.87 | 0.6 | 2.07 | 0.9 | 3.78 | 50.41 |

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
