# Peer review of "A Shift in Communities of Conspicuous Macrocrustaceans Associated with Caribbean Coral Reefs following A Series of Environmental Stressors"

_diversity, doi:10.3390/d15070809_

Round 1
Reviewer 1 Report
This study compares the crustacean motile at two Mexican Caribbean localities before and after a series of environmental disturbances. The ms is clearly written and illustrated. I have a few comments that may perhaps be help the authors to clarify some matters.
Major concern, It is true that motile crustaceans constitute a large part of the reef-associated invertebrate fauna (Line 74), but there is another component, coral-associated crustaceans, that is not mentioned and that may be even more affected by disturbances. These can be like for instance coral barnacles and coral gall crabs that live partly inside the corals. Coral degradation will have an immediate effect on these crustaceans. It may be helpful to the readers if you mention (in the introduction or discussion) their importance for coral reef biodiversity but that these were not included because they are usually small and cryptic. Their role in coral reef biodiversity has been reported last year in two separate studies from Sint Eustatius in the eastern Caribbean (Lymperaki, Ecological Engineering 176:106536) and from Curacao in the southern Caribbean (van der Schoot, Marine Environmental Research 181:105738). The relevance of host corals is also illustrated by Domecia acantophora in your own study since this species also has dwellings in corals [66,67]
Detailed comments
Line 61. Since the 1980’s -> Since the 1980s
Line 351. Do not start a sentence with a genus name abbreviation: M. coryphe -> Mithraculus coryphe
Appendices. The shrimp Thor amboinensis is not a Caribbean species. This should be Thor dicaprio Anker & Baeza, 2021. See: Anker, A.; Baeza, J. A. (2021). Thor dicaprio sp. nov., a new, conspicuously coloured shrimp from the tropical western Atlantic, with taxonomic remarks on the T. amboinensis (De Man, 1888) complex (Decapoda: Caridea: Thoridae). Zootaxa 5039(4): 495-517.
Appendices. I am missing the crab Platypodiella spectabilis, which is hosted by zoantharians but can also be motile (Garcia-Hernandez et al. 2016, Coral Reefs 35:209). Do you have an explanation for this? Is it absent or has it been overlooked?
Reviewer 2 Report
Major comments: The manuscript 2347092 entitled “A shift in crustacean communities associated with Caribbean coral reefs following a series of environmental stressors” submitted to Diversity by Melissa Dubé is both exiting and problematic. Firstly, I find this study highly interesting for several reasons: (i) it is comparative in referring to an earlier study, (ii) it raises methodological questions (reef rugosity), (iii) it provides a striking example for an increase in species richness (positive) that might come at the cost of reef degradation (negative) and (iv) it illustrates the complexity of stressors to which ecosystems are exposed, resulting in equally complex ecosystem changes, thus highlighting the difficulties in the interpretation of causal relationships between stressors and changes. Secondly, the study is problematic in that (a) it makes inferences that are not fully supported by the data or, more precisely by the replications, (b) uses a method (reef rugosity) that, as the authors recognise themselves, is apparently not pertinent or sensitive enough to reveal differences in reefs that are visibly quite different and (c) does not explain sufficiently why an increase in biodiversity could be considered a negative outcome. In short, I would recommend publication in Diversity because of the interesting results, which can be compared to an earlier study, but would recommend major revisions with respect to the inferences made from the data. Most of all, it is advised to be more conservative in the interpretation of the results and to clearly highlight the limits of the study.
My central criticism to the study is that it compares two datasets, each of which represents one value for a given parameter with its inherent variability. Although within each dataset the are several samples that constitute the means, the overall sample number is two (n=2), corresponding to the first and the second study, or within each study to the one and the other reef. On the other hand, each n is subjected to multiple variables that may, or may not modify the observed means. Statistically, you can rank the values for each dataset and compare them by a Mann-Whitney U test, but with respect to the inferences which environmental factor is responsible for any observed statistically significant difference it is obviously very difficult to draw conclusions on such a data basis. The authors might object that such studies are laborious and costly. But the only way to obtain more robust data is to increase replication and to conduct a meta-analysis. This is also done in other disciplines and may allow for the identification of factors that do or do not contribute to the variability over time. An information-theory model selection approach based on Akaike's Information Criterion could perhaps be useful using the species and microhabitat data, but it will be difficult to test hypotheses as to whether SCTLD, Sargassum, hurricanes or other factors are responsible for variability in the observed data (unless the authors have precise data to feed the models with). For the study as it is, this does not prevent publication, but the authors need to be clear about the limitations that are imposed on the interpretation of the results. It must be acknowledged that some of the speculations in the discussion are plausible and well argued, whereas others are excessively speculative and have little support. To my humble point of view, what is necessary to render the manuscript publishable is to make the speculative nature of many of the statements in the discussion very clear to the reader and refrain from some interpretations that are hardly supported. It is understood that we all make observations and that we try to give reason to these observations. In science this must lead to testable hypotheses, which are then verified or falsified with a sufficient number of replicates. For the most part this is not the case in the present study (only the for decapod communities, but not for any of the stressors that may be held responsible for the differences). The inferences must, therefore, be preliminary in nature.
Detailed comments:
Title: why “crustacean communities” if it is only decapods (see comment on the methods section below)? This raises expectations that are not fulfilled. “A shift in decapod communities …” would be more correct, wouldn’t it?
Abstract: I might be completely mistaken, but it is stated that the effect of the stressors can be evaluated by measuring the changes in the habitat and the communities of reef-associated crustaceans. How can you, by comparing the state before and after the events identify the cause for the changes? As the authors state themselves in l. 432-433, the contribution of the stressors cannot be disentangled. I think that this is inherent to the ‘study design’. It might suffice to rephrase the objectives more conservatively. You just establish the state before and after the events and provide plausible explanations for the observed ecosystem changes (indeed, the authors are doing a very good job in providing fairly sensible arguments for the hurricanes being mainly responsible for the observed shifts in crustacean community composition). The last sentence of the abstract is perfectly phrased in this regard, saying that the results “suggest” …
Introduction: To my liking the introduction is not sufficiently detailed (as opposed to the discussion). Some statements would require more tangible examples, others should be better explained.
First paragraph: “… important ecological services …”. Like what? “… decrease in seawater quality …”. Vague! Decrease in water quality can mean a lot. Please specify!
The problem of the massive influxes of pelagic Sargassum has to be explained, as it is not straightforward. Sargassum needs nutrients to grow and to maintain themselves. Since they grow in Sargassum belt, they are likely to absorb most nutrients there. It should be made clear that these nutrients are then transferred, and when the Sargassum becomes senescent and eventually dies, the nutrients are released and add to the anthropic eutrophication from the coast. Besides, it is the decrease in dissolved oxygen by the degradation of algal biomass that causes problems and massive death.
Seagrasses (l. 56) are another component that is important for biodiversity. Even if this ecosystem was not studied, it was recognised as a habitat. Have the authors any idea how it was affected by the hurricanes. What interactions exists in terms of biodiversity between the seagrass meadows and the coral reefs? It is a pity that the seagrass meadows have received little attention in the study. They are one of the microhabitats included figure 6 that seem to change between 2015 and 2022.
As for the stony coral tissue loss disease, where did it originate from, when has it become prevalent and has the SCTLD been attacking the corals more ferociously because the corals have been weekend by other stressors, for instance those mentioned above?
l. 72: Why is spanning a broad size range so important?
Why are crustaceans so important? Perhaps one argument that the authors may put forward is that the studied decapods have different feeding strategies and are found at different positions in the food web.
l. 76: How can these differences be explained? Does it have to do with the invertebrate taxa were studied? But not only the differences in results are important to be explained. Also, it should be made clear that an increase in biodiversity does not always have to be a positive result. As the authors explain in their discussion, reef degradation may entail any increase in species richness for the studied decapods, but at the same time specific symbiotic coral-crustacean relationships might be lost. Other invertebrates might decrease in number if predating decapods exert pressure on them. A more differentiated perspective would be good to have here.
Last paragraph: I find this paragraph rather chaotic. Perhaps first introduce the study then its results and then fact that was repeated and the reasons for it. The citation (l. 84-88) is not needed.
Material and Methods:
first paragraph: Many other factors could be responsible for the differences (exposure to currents, food availability, nutrients for the algae etc.)! The authors suggest that tourism is the principal factor, but do not provide proof. Also, they do not mention this factor elsewhere amongst the factors the effects of which they want to study (SCTLD, Sargassum, hurricanes). Obviously, the position of the reefs is not the same with Limones being the outermost reef. What is with the other reefs?
l. 109-112: here important information on the live coral cover is provided, but as far as I am aware of such information was not used in the evaluation of the reefs. I understood that for evaluating the health of the coral reefs, only rugosity was used.
2.2. Macrocrustacean surveys:
I have difficulties to imagine, how the divers did bring up small crustaceans hiding under algae, stones, in the sand or in the gravel? Most crustaceans strongly react to light changes brought about by shade and movement. A diver is thus likely to chase many animals away. Please explain in more detail how it was assured that the inventory was complete.
How many divers were analysing the transects independently? In l. 137 and l. 138 it is stated that “… two scientific observers …” “… and a third observer …” identified the species. But were the transects inspected independently or did the divers work together (in other terms, do you have three values for the species inventory or only one?)? This is unclear, especially by l. 141 “… the results were cross checked between divers.”, which suggest that there were several results. Were the divers the same as in 2015? Please clarify these aspects.
l. 132: I was wondering about the absence of orders like isopoda, which could count amongst the macrocrustaceans. I recognised that all taxonomic groups comprise infraorders belonging to decapoda. It would perhaps be helpful to replace “crustaceans” by “decapods” reading as follows “… defined as motile decapods larger than ~1.5 cm …” to avoid any misunderstandings. You may also do so throughout the manuscript, where either the term macrocrustaceans or just crustceans are used. I think that these taxa are too broad and somewhat misleading.
l. 141: In the comparison (cross-check), what was the percentage of mis-identifications? If different identifications happened to occur, what was the procedure to deal with these “conflictual” identifications?
2.3. Reef rugosity: I am not an expert on measuring reef structure and the authors have certainly used a common method. Notwithstanding, I’d like to ask about the pertinence of this method, which does not show any difference, despite the reefs seem to be functionally very different. The authors conclude themselves that “standard metrics of reef condition such as […] structural complexity” (l. 308-309) do not capture the differences.
I figure, the method measures the structure build by the corals over some time. If corals die, how long would the authors estimate may it take to see a difference between a reef with living - and still building - corals and a more or less dead reef. The authors confirm that dead reefs flatten and loose structure. But seven years is likely to be insufficient to detect differences in rugosity, is it?
In figure 2, I count approximately 20 points per reef and year. Does this correspond to 20 transects? There were 30 transects analysed for the decapod crustaceans. Was there any difference between the number of transects? Please specify how much transects for rugosity were measured per reef!
2.4. Data analysis: I think the study has a major drawback in being ‘undercomplex’. Basically, the authors have measured rugosity, macrocrustacean community and characterised the associated habitats. Rugosity does not seem to be an adequate measure of reef degradation and because of that the study reports parameters on macrocrustacean community, but is unable to correlate them with any factor that might have modified these communities. The authors aim to evaluate the effects of various stressors such as SCTLD, increased nutrient load by Sargassum (possibly including oxygen decrease by degrading algae) and hurricanes, but they do not present any data that allow for the quantification of these effects and how they would have impacted the reefs differently. If the authors could procure related data and/or perhaps other data that distinguish the two reefs (perhaps from some external databases), these data might be correlated with the crustacean community data, and explain the variability in the data.
General comment on the methods: The authors qualify the methods as “standardised” and without any doubt they have taken great care to assure that the investigators were well formed and instructed to carry out the observations. Nevertheless, the acquisition of the datasets depends on individual observations, which are likely to include some individual observational plasticity. The authors cannot exclude that this ‘investigator-plasticity’ accounts for some of the data variability between 2015 and 2022. The emphasis on ‘standardisation’ is suggestive of ‘absolute’ values that are fully comparable, such as if a spectrophotometer measures light absorption. But this is not the case. Perhaps the authors should discuss this aspect in a more differentiated manner.
Results:
3.1. Reef rugosity: In 2015, reef rugosity for both sites had higher variability. It cannot be excluded that investigator-variability may account for some of the variability. It is also possible that the environmental stressors, such as hurricanes, or any other factor may be responsible for the changes. But how can anybody tell? Even if I agree with the authors that the datasets provide evidence for reef degradation of Limones (especially with respect to the microhabitats in figure 6), the authors should point to some variability that may have been introduced by observational bias.
3.3. Microhabitat use by macrocrustaceans: I feel that what is lacking here is a quantification of the surfaces that each microhabitat comprises in both reefs. Have the authors any data on the surfaces covered by corals, algae, dead corals and rubble (by the way, what is the difference?), seagrass meadows etc.? This could help to better interpret the community data.
l. 208 ff: The authors present the distribution of several infraorders on the two reefs in 2015 and 2022. As far as I am aware of, the authors do not inform about whether any of the differences between reefs and between years would be significant. There are statistical procedures that allow to compare proportional data. For instance, the raw data can be compared with a chi-square test. Other statistical tests can also be adapted to comparing proportions.
l. 273: same comment as for infraorders. It seems that there clearly are significant differences between the reefs in terms of crustaceans occupying live corals as well as dead corals and rubble or algae and most likely there are significant differences between the years in the proportions that depend on life corals as habitats. Nevertheless, one would like to know whether the observed differences are true differences, i.e., statistically significant. This can be calculated.
Discussion:
l. 289-290: I do agree with the authors, but as pointed out before, even if standardised, the methods are essentially observational ones. It is known that such methods are influenced by individual bias between observers (as, luckily, human beings are not yet standardised). In addition, each observer has prior knowledge about the reefs that are compared. The analyses are, therefore, not ‘blinded’. In some other disciplines that make use of observations, the observations have to be blinded and repeated by different analysts. It is acknowledged that the analysis has been ‘repeated’ 30 times (corresponding to 30 transects) and this is probably as good as can be done. At the same time, how many divers were analysing each transect indepently and were they the same as in 2015? Can the authors exclude any plasticity between the observations made in 2015 and 2022? Probably not! It will be difficult to quantify the variability introduced by observational plasticity, but the authors should discuss that at least the possibility exists that observational bias might account for some of the variability in the data.
l. 306-309 and l. 316-317 ff: I would prefer that the authors clearly question the pertinence of the “standard metrics of reef condition”, especially rugosity, which stands for “reef complexity” and suggest other methods able to qualify the differences between the reefs (except for decapod community).
Besides “… the crustacean communities …” appears too large; only decapods have been analysed (see comments above).
l. 346: Sargassum is mentioned in the abstract and introduction as a factor that might have modified the reefs, especially Limones, but nowhere has it been explained why it would affect the reefs differently. Furthermore, the authors have not explained what would be the effect of Sargassum on the reefs and the decapod communities. In Bonanza the algae are probably promoted by the additional nutriments, although it could be that the waters are already as rich in nutrients (degradation of water quality mentioned in the introdcution) from coastal runoff that additional nutrients make little difference; in Limones the nutrients from Sargassum may increase algal overgrowth - again one would need data.
l. 394 and l. 400-401: This is where to my point of view the problem lies! The authors observe a phenomenon once and want to explain it: “so the aggregations of C. tricolor may have been favored by a local increase in detritus”. This is pure speculation not supported by any data and should be removed.
l. 433: Perhaps the authors can critically discuss why the observed changes cannot be disentangled and outline some improvements to the protocol necessary to be able to disentangle changes observed by reef-monitoring in the future, so as to avoid the problems encountered in this study.
l. 452-453: Please rephrase in order to be less affirmative! It is of course probable and even likely that the events have impacted the reefs, but the authors cannot exclude that other factors may have influenced the results. Hence, “… a series of events … have brought about changes …” is suggestive. More reservation with respect to the causal relationshipds would be indicated.
Conclusion: Much of the conclusion is repetitive and has been formulated identically elsewhere in the manuscript.
Figures: figures 3 and 6 do not indicate whether the proportions are significantly different. If it is too complicated to include this information visually in the figures, the authors may add a table listing the probabilities.
Typos and other corrections:
l. 14: better “healthy” instead of good
l. 44-45: “… increased in these coastal ecosystems, …”
l. 46: “… were being affected …”
l. 73: “… of how such recent phenomena …”
l. 74-74: perhaps “Decapods constitute an important part of the reef-associated invertebrate fauna …”
l. 92: “Hurricanes contribute substantially/importantly to reef degradation …”
l. 103: “… is a protected marine area …”
Figure 6: Other live corals – a “r” is missing
Table B1 – B) Limones 2022: please replace “Especie” by Species.
As far as I can judge, the English is of high level and good to read. Only very few corrections are necessary.
Round 2
Reviewer 2 Report
I would like to congratulate the authors for their work and for the changes they have made to the manuscript so quickly. I do believe it has increased in pertinence, but this is, of course, only my humble opinion. If the authors do feel the same, then I might have made a small contribution by reviewing this excellent study.
The authors have responded to all of my comments and I do not have further amendments to ask.
I would just like to comment on the following: It is of course known to me and agreed upon that evaluating the causes for reef degradation is difficult as they are multifactorial and certainly also because the research group has limited capacities to collect all the data that would be necessary. It is also clear to me that field work differs from laboratory experiments in its possibilities to control confounding factors (and even in the lab this is not a hundred percent possible). I do think, however, that it is important to render these difficulties transparent, especially to interested researchers, as I am, that do rather come from a laboratory environment. Furthermore, I believe it would be helpful, to demonstrate to regulatory authorities and politics what means would be required to gain more knowledge and more certainty about the causes for reef (and other) degradation.
Once more I would like to compliment the authors for this follow up study, which is particularly compelling as it allows to demonstrate that a) the reefs are not getting better and b) the authors have the tools and the knowledge to demonstrate this. Even if one cannot be fully sure about the contribution of each stressor to this degradation, it is plausible that the environmental conditions the authors bring forward, do in fact contribute to the degradation.
However, this is where the problem lies: A scientific proof is lacking and until this cannot be provided, politics and legislators will tend to dispute the conclusions. I, therefore, think that it is important to upscale this kind of study by providing data that allow for correlations between the biodiversity data, reef degradation and environmental factors in order to demonstrate that the assumptions made are indeed correct. This will be difficult, and in complex ecosystems it will be hard, if not impossible, to take into account all factors and the interactions between them. Nevertheless, scientists should strive for advancing the situation as much as they can (which the authors certainly did!).
Perhaps, and if I still might suggest something, it would be interesting to give an outlook on how to improve the data basis in order to have more reliable information about the causes for the reef degradation. Also, the authors could point to the resilience of the reefs, which as yet has kept changes minor, but which is not guaranteed in the long-term.
Another point, which is intriguing, and which might be commented on: increased biodiversity usually is considered positive. But here it comes at a price of loosing precious coral-associated symbioses and it might also imply other disturbances within the ecosystem that have not been analyzed.
